# Investigating the Difference in Factors Contributing to the Likelihood of Motorcyclist Fatalities in Single Motorcycle and Multiple Vehicle Crashes

**DOI:** 10.3390/ijerph19148411

**Published:** 2022-07-09

**Authors:** Ming-Heng Wang

**Affiliations:** Department of Traffic Management, Taiwan Police College, Taipei 11696, Taiwan; wang.mingheng@gmail.com; Tel.: +886-978-283-257

**Keywords:** vehicle involvement, likelihood of motorcyclist, subjected motorcyclists, counterpart motorcyclist, blood alcohol content (BAC)

## Abstract

In order to better understand the factors affecting the likelihood of motorcyclists’ fatal injuries, motorcycle-involved crashes were investigated based on the involvement of the following vehicles: single motorcycle (SM), multiple motorcycles (MM) and motorcycle versus vehicle (MV) crashes. Method: Binary logit and mixed logit models that consider the heterogeneity of parameters were applied to identify the critical factors that increase the likelihood of motorcyclist fatality. Results: Mixed logit models were found to have better fitting performances. Factors that increase the likelihood of motorcyclist fatality include lanes separated by traffic islands, male motorcyclists, and riding with BAC values of less than the legally limited value. Collisions with trees or utility poles lead to the highest likelihood of fatality in SM crashes. The effects of curved roads, same-direction swipe crashes, youth, and unlicensed motorcyclists are only significant in the likelihood of fatality in SM crashes. Conclusions: Motorcyclists tend to be killed if they collide with large engine-size motorcycles and vehicles, unlicensed motorcyclists, or drivers with speeding related or right-of-way violations with positive BAC values. Driving or riding should be prohibited for any amount of alcohol or for anyone with a positive BAC value. Law enforcement should focus on unlicensed, speeding motorcyclists and drivers, and those who violate the right of way or perform improper turns. Roadside objects and facilities should be checked for appropriate placement and be equipped with reflective devices or injury protection facilities.

## 1. Introduction

According to the Global Status Report on Road Safety (WHO, 2018), approximately 1.35 million people die every year in road traffic crashes. More than half of road traffic deaths are among vulnerable road users including pedestrians, cyclists, and motorcyclists. Pedestrians and cyclists represent 26% of all deaths, while motorcyclists who use motorized two- and three-wheelers represent a further 28%. To address motorcycle safety issues, numerous studies have been conducted to investigate the critical risk factors of crash occurrence and injury severity in motorcycle-related crashes [1,2,3,4,5,6,7,8,9,10,11,12,13,14,15,16,17,18,19,20,21,22,23,24]. However, most of them were conducted under limited motorcycle traffic with mostly large motorcycle engine sizes (more than 250 cc), a limited sample size of crash cases, and without a restricted road deployment for motorcycles. For example, a study that analyzed injury severity outcomes for single motorcycle crashes used only 3644 cases [9].

In Taiwan, there were more than 14 million registered motorcycles by the end of 2021. About 98.6% of them had engine sizes of less than 250 cc. Most roads deploy motorcycle-exclusive or motorcycle-priority lanes, some of which are separated by traffic islands to decrease the interaction and conflict between motor vehicles and motorcycles. Motorcycles are also restricted from traveling on freeways and expressways. On most streets, if motorcycle or nonmotor lanes are deployed, motorcycles with an engine size of less than 250 cc can only travel on the motorcycle or nonmotor lanes and the nearby vehicle lane. However, motorcyclists still make up the largest portion of traffic crash injuries and fatalities. In 2020, more than 85% of traffic injuries were motorcyclists or riders, and more than 64% of all traffic fatalities were motorcyclists or riders.

Although previous studies have found numerous risk factors contributing to motorcycle crashes and injury severity, it is still important to investigate the critical contributors which affect motorcyclists’ injury severity under a particular motorcycle management environment with large amounts of motorcycle traffic. While previous studies mainly focused on motorcycle crashes with other motor vehicles, only a few of them dealt with single-motorcycle (SM) crashes separately [9,10,11,12]. This is despite the fact that about 28% of motorcyclists die in single-motorcycle crashes. Previous research has asserted that single- and multi-vehicle crashes can be best described when modeled separately [13]. However, none of them consider the crashes between motorcycles only separately. Due to the size and physical characteristics of motorcycles, the injury severity of motorcyclists would be different in collisions between only motorcycles (motorcycle vs. motorcycle, MM) and collisions with other motor vehicles (motorcycle vs. vehicle, MV).

Furthermore, in multiple-vehicle crashes, the characteristics of counterpart motorcyclists or vehicles (collision partners) are rarely addressed in previous studies beyond vehicle type. Thus, the factors contributing to a motorcyclist’s fatality need to be investigated separately with regard to the number of motorcycles and vehicles involved. The characteristics of counterpart motorcyclists and vehicle drivers in MM and MV crashes were also included in the analysis.

In terms of the methodology, several works demonstrated that logit models are effective tools for exploring the factors that influence the injury severity of traffic crashes [25,26,27,28]. However, to address heterogeneity among drivers, several recent studies used a variety of statistical approaches to identify the effects of unobserved factors on the injury severity of crashes. Methods include heteroscedastic logit models, random effect models, latent class models, random parameter logit models, and a random parameter approach with heterogeneity in the means and variances [29,30,31,32,33,34]. In addition to the binary logit model, this study also introduced random parameters to fit the crash data.

The objective of this study was to investigate the factors contributing to motorcyclists’ fatalities in motorcycle-related crashes. Five years (2016–2020) of police-reported, motorcycle-related crash data were divided into the following three subsets: single-motorcycle, motorcycle–motorcycle, and motorcycle–vehicle crashes. Descriptive statistics of the crash data, such as sample size, distributions of variables including road environment, crash type, human and vehicle characteristics, are presented. Both binary logit and mixed logit models that consider the heterogeneity of parameters were conducted using three datasets to identify the fixed and random effects of risk factors which increase the likelihood of motorcyclist fatality. The probability of motorcyclist fatality was estimated to examine the related impact of those critical factors. Furthermore, the fitting performances between different models were compared and discussed to confirm critical factors that may increase the risk of fatality for motorcyclists. The conclusions drawn provide a helpful reference for related research on motorcycle safety and crash prevention.

## 2. Related Works

Motorcyclists are vulnerable road users who tend to suffer severe injury due to the lack of protection in traffic crashes [5]. Several studies have been conducted to investigate the contributing factors to the severity of motorcyclists [1,2,3,5,6]. Vlahogianni et al. [22] reviewed and discussed critical factors affecting motorcycle safety, including crash occurrences and injury severity. They divided the critical factors into the following four categories: motorcyclist behavior, road infrastructure-related, vehicle-related, and weather-related risk factors. Yousifac et al. [35] also reviewed research regarding the impact of behavioral issues on motorcycle safety. They summarized and discussed several behaviors, such as speed, visibility, fatigue, and drowsiness, which lead to an increase in the probability of motorcycle crashes and severe injuries. Other studies focused on identifying factors such as the road environment and human and vehicle characteristics on the injury severity of motorcycle crashes [7,8].

Regarding motorcyclist characteristics, age and gender are among the main factors that affect motorcyclists’ injury severity in the crashes. The likelihood of fatality and disabling injuries in single-vehicle motorcycle crashes increases with motorcyclist age [9,10,14,15,16]. Older motorcyclists are more likely to suffer severe injuries because of their slower reaction time and lack of sensory and perceptual ability, as well as their decreased physical resilience to crashes when compared to younger drivers [10,15]. Previous results on the effects of gender on injury severity were varied. Savolainen and Mannering [10] found that female motorcyclists are more likely to be involved in non-capacitating injury crashes. However, Shaheed and Gkritza [11] concluded that a male motorcyclist involved in a single-vehicle motorcycle crash was more likely to be less severely injured when compared to a female motorcyclist.

Alcohol is an important risk factor of motorcycle crashes directly related to the decrease in riding skills and attention [2,17,36,37,38], and may be associated with speeding, non-use of helmets and unlicensed riding [39]. However, none of the studies discussed the effect of BAC values, especially within the legally accepted values, on the motorcyclist injury severity.

Helmet use was found to reduce the risk of motorcyclist fatalities [17,18,19]. Riding a motorcycle without a license [20] increases the severity of motorcycle crashes. Past studies have also shown that motorcycle engine size is associated with motorcycle crash injury severity outcomes [7,10,15,16,21].

Road type and geometry, along with roadside installations, pavement surface conditions, lighting, visibility conditions, and manner of crashes (such as run-off-road and collision with a stationary object) can also influence crash outcome [22]. With regard to lighting and visibility conditions, poor visibility due to horizontal or vertical curvature or darkness has been associated with an increase in motorcycle injury severity [10,11].

Riding on a wet roadway surface can be a risk factor as well. However, the crash risk may decrease if motorcyclists maintain a lower speed due to caution toward poor surface conditions [10].

Regarding the crash configuration (type of collision), research has shown that collisions with stationary objects result in more severe injuries [10,11,16], while types of stationary objects were not analyzed. The impact of the utility pole and traffic island should be different due to the differences in their contact area during the collision.

Collisions with heavy vehicles [23] and riding a motorcycle during midnight or early morning [24] can also increase the severity of motorcycle crashes.

## 3. Data Preparation

The data used in this study were extracted from the National Police Administration (NPA) traffic crash database. The NPA crash data are police-reported data including crashes where at least one road user is killed or injured. Variables recorded in the NPA crash database include the characteristics of crashes, road environments, vehicles, and humans (drivers, motorcyclists, passengers, and pedestrians or others related to the crashes). The following three types of motorcycles were classified in the NPA database: Large heavy motorcycles (LHM—engine size 250 cc (cubic centimeters) and above), general heavy motorcycles (GHM—engine size between 50 cc and 250 cc), and light motorcycles (LM—engine size of 50 cc).

Crash characteristics include the year, month, day of the week, time of day, and crash types. Characteristics of the road environment included the types of land use (rural or urban), roadway alignment and profile, traffic control devices, lighting conditions, traveling lane types, and separation facilities. The roadway alignment profile is grouped into the following three categories: intersection, curve (vertical or horizontal) grade, and straight segments. A particular road alignment in Taiwan is that most major roads deploy motorcycle lanes or nonmotor lanes. Most of them are separated by markings, while some of them are divided by traffic islands or barriers which restrict motorcycles with engine sizes of less than 250 cc.

The characteristics of vehicles and humans include vehicle type, motorcyclist’s or driver’s sex, age, license validity, blood alcohol content (BAC), helmet/seat belt usage, and inappropriate behaviors or violations in the crashes. In some cases, motorcyclists or drivers may have multiple violations but only the one that is the major contributor to the crashes is recorded in the NPA dataset. For example, if a motorcyclist is impaired and has a right-of-way violation, the major cause of the crash goes to the right-of-way violation. The motorcyclist or drivers’ violations and inappropriate behaviors were determined by expertly trained officers and recorded in the NPA database.

The motorcyclist or driver’s BAC tested values were recorded in five scales with various ranges of BAC values in the NPA database. According to the Taiwanese traffic regulations, drivers or motorcyclists are considered impaired if their BAC results are 0.03% g/dL or above. To understand the effect of BAC values on the likelihood of motorcyclist fatality, BAC values are recorded in four scales, zero, less than 0.3% g/dL, larger than 0.3% g/dL, and “unknown”.

Crash types are categorized into several major types based on crash involvements. In single-motorcycle crashes, major crash types include run-off, hitting traffic devices, such as signal or sign poles, hitting trees or utility poles, and hitting the traffic island or barrier. In multiple motorcycle or vehicle crashes, major crash types are head-on collision, swipe crash in the same or opposite direction, rear-end collision, side-impact, and right-angle collision.

The data used in this study consist of five years (from 2016 to 2020) of crashes involving one or two motorcyclists. The motorcyclist-involved crash data are divided into the following three datasets for this study: crashes involving a single motorcycle (SM), two motorcycles (MM), and one motorcycle and one motor vehicle (MV). In single motorcycle crashes, crashes involving other vehicles due to secondary consequences of the first crashes were excluded from the analysis. For example, crashes in which a vehicle hits the barrier first, then collides with another vehicle were excluded. In multiple vehicle crashes, crashes involving more than two vehicles were excluded from the analysis because one of the vehicles might have been a consequence of the first collision. The crash dataset did not report which of the two involved vehicles was responsible for the first collision.

Crashes involving bicycles, pedestrians, and other non-motor vehicles were also excluded from the analysis. Crashes occurring on the freeway and expressway were also removed from the data. In Taiwan, all motorcycles regardless of size are prohibited from traveling on freeway systems, and only LHMs are allowed to travel on expressways.

A total of 122,710 motorcyclists in SM, 771,029 motorcyclists in MM, and 735,270 motorcyclists in MV crashes were used in the final dataset for analysis. Each crash data point records the injury severity (fatal or non-fatal) of the motorcyclists involved in the crash, along with human, road environment, and crash characteristics. Characteristics regarding the crash, road environment, motorcyclists, and the fatality frequency and percentage distribution for these crashes are listed in Table 1. Note that the human characteristics include age, gender, improper behavior or violations, license, and the alcohol consuming statuses of subjected motorcyclists and counterpart motorcyclists and drivers (collision partners). The engine size of the motorcycle and vehicle type are also included.

## 4. Methods

In this study, two analysis methods, binary logit model and mixed logit model, were introduced to investigate the majority of factors influencing the likelihood of motorcyclist fatality. The results from these two methods were compared and discussed.

### 4.1. Binary Logit Model

A logistic regression analysis was conducted to examine the relevant variables for determining the odds of motorcyclist fatality. Logistic regression was applied due to its suitability for predicting binary dependent variables as a function of predictor variables. Logistic regression models have been widely used in road safety studies involving a binary dependent variable [25,26,27,28]. The binary logit model framework is used to model the likelihood of fatal injuries sustained by a crash that involves a motorcyclist, as represented in Equation (1)
(1)$$F_{i}=\beta X_{i}+\varepsilon_{i}$$
where *F_i_* is a linear function that determines fatal injury for motorcyclist *i*. β is the vector of coefficient estimates, *X_i_* is the vector of characteristics (driver, vehicle, and environmental attributes) that influence a motorcyclist’s fatal injury, and *ε_i_* is an independent and identically distributed generalized extreme value (i.e., Gumbel type 1) error term. Each motorcyclist has the same set of potentially fatal injuries and the probability of a crash resulting in fatal injury may be found from Equation (2):(2)$$P_{i}=\frac{\exp\left(\beta X_{i}\right)}{1+\exp\left(\beta X_{i}\right)}$$
where *P_n_* is the probability that motorcyclist *i* will sustain fatal injuries.

The stepwise logistic regression by R [40] checks the multicollinearity among variables. Independent variables with high correlations are removed from the final model. It then performs model selection using AIC (Akaike Information Criterion) with the backward deletion technique.

### 4.2. Random Parameter Mixed Logit Model

Due to the limitations of data recorded from the traffic crash scene, several important pieces of information that may affect a motorcyclist’s injury severity were missing. For example, a motorcyclist’s physical and mental status (e.g., elderly motorists’ physical impairments or cognitive deficiencies) and their risk perception (risky lifestyle, aggressive behavior) have been reported to contribute to motorists’ violations in motorcycle crashes [4], but this information is not available in the crash database. Thus, a mixed logit or random parameter model that considers the unobserved heterogeneity issue was introduced. The mixed logit model considers that the parameters may vary across observations, indicating that the random parameter’s coefficients may vary from the fatal injury of one motorcyclist to another motorcyclist involved in crashes.

The random parameter mixed logit model can be constructed as shown in Equation (3):(3)$$F_{i}=\beta X_{i}+\left(\eta_{i}+\mu_{i}\right)$$
where *F_i_* is the value of the function that determines the fatal injury for motorcyclist *i*, β is the vector of coefficient estimates, *X_i_* is the vector of non-stochastic variables which affect fatal injury, *η_i_* is the random term with zero mean, and *ε_i_* is the error term that is independent and identically distributed and does not depend on underlying parameters or data [41].

The mixed logit model is a generalization of the binary logit structure that allows the parameter vector β to vary across individuals. The outcome-specific constants and each element of β may be either fixed or randomly distributed across all parameters with fixed means, allowing for heterogeneity within the observed crash dataset. A mixing distribution was introduced to the model formulation, resulting in injury severity probabilities as shown in Equation (4) [42]:(4)$$p_{i}=\int \frac{\exp\left(\beta X_{i}\right)}{1+\exp\left(\beta X_{i}\right)}f\left(\beta |\varphi \right)d\beta$$
where f(β|φ) is a density function of *β* and *φ*; *β* is a vector of the parameters which describe the density function; and *φ* is the vector of the random parameter means and standard deviations. The fatal injury probability is then simply a mixture of logits [41]. The *β* distribution may allow for individual-level variations of the effects of *X* on the resultant fatal injury. The distribution is also flexible in that *β* may also be fixed, and when all parameters are fixed, the model reduces the standard binary logit formulation. In instances where *β* is allowed to vary, the model is in open form and the probability of a motorcyclist sustaining a fatal injury can be calculated through integration.

The random parameter logit model assumes that the corresponding coefficients of the explanatory variables can obey a certain distribution. Referring to previous studies, the random parameter was also assumed to be normal distribution in this study [43].

Estimates of the mixed logit model by maximum likelihood are processed using simulation approaches due to the difficulty in computing the probabilities [43]. Halton draws are considered the most widely used method and significantly more efficient than purely random draws [43]. Initially, all variables are included in the model, and then backward elimination is employed to remove parameters that are not significant at the 95% confidence level. For the random parameter mixed logit model, 400 Halton draws were drawn from simulated normal distributions. Random parameters were identified as parameters with standard deviations different from zero at the 95% confidence interval.

To avoid any multi-collinearity between explanatory variables, the variance inflation factor (VIF) was tested for each developed model. Variables with a VIF value of greater than 10 were removed from the final models.

### 4.3. Likelihood Ratio Test

Upon development of the binary logit and mixed logit models of fatal injury for all motorcyclist-involved crashes, the likelihood ratio test was used to determine whether those two models were different or not. The null hypothesis of this test is that there is no statistical difference between the binary logit model and the model with random parameters. The likelihood ratio test statistic follows chi-square distribution and is presented as in Equation (5) [43]:(5)$$\chi^{2}=-2\left[LL(\beta_{Binary}\right)-LL(\beta_{Mixed})]$$
where *χ^2^* is the test statistic, $LL(\beta_{Binary})$ is the Log-likelihood at the convergence of the binary logit model, and $LL(\beta_{Mixed})$ is the Log-likelihood at the convergence of the mixed logit model, where parameters are free to vary among two models. The resulting $\chi^{2}$ test statistic has degrees of freedom equal to the difference in the parameters between the two models, which is equal to the number of random parameters.

### 4.4. Impact of Parameters

To comparatively assess the impacts of each parameter, odds ratios were also estimated. The odds of an event are defined as the probability of the outcome event occurring divided by the probability of the event not occurring. An odds ratio that is equal to exp(*β*) provides the relative amount by which the odds of the outcome increase (OR greater than 1.0) or decrease (OR less than 1.0) when the value of the predictor value increases by 1.0 units [44]. The odds ratios represent the risk of fatality comparison among different levels [29,30].

The ORs are presented in the developed models. When the independent variable *X_i_* increases by one unit (with all other factors remaining constant), the odds increase by a factor of OR, which ranges from zero to positive infinity. The OR indicates a relative increase (risk of fatality) (OR > 1) or decrease (OR < 1) in the odds of the outcome when the value of the corresponding independent variable increases by one unit.

## 5. Results and Discussions

Based on the crash datasets, binary logit models and heteroscedastic logit models for SM, MM, and MV crashes were developed, and the estimated results are presented in Table 1, Table 2 and Table 3. For all cases, the mixed logit model exhibits a better fit than the binary logit models with a low AIC value and higher McFadden R2. The odds ratios obtained from the mixed binary are presented in Table 4.

### 5.1. Random Parameters

#### 5.1.1. Single Motorcycle Crashes

The modeling results based on both binary logit and mixed logit models for the single motorcycle crashes are shown in Table 2.

The results find that the AIC value in the mixed logit model (12,138) is less than that in the binary logit model (12,151). The Chi-square (*X*^2^) statistical value and the corresponding degree of freedom, 5 (the number of random parameters), finds that the calculated value (13.19) is greater than the critical value (11.07) at the 95% confidence level. Thus, the null hypothesis is rejected, which again proves that the random parameter logit model is superior to the binary logit model.

As shown in Table 2, the following five parameters were identified as random parameters with statistically significant standard deviations: crashes at midnight, hit traffic devices, motorcyclists aged 55 years old or above, motorcyclists with a BAC value of less than 0.03%, and motorcyclists with a BAC value of more than 0.03%. It is noted that when the random effect of the parameters is considered, some factors on motorcyclist fatality turn out to be insignificant. For example, the impact of motorcycle engine size on fatality risk is not significant in the mixed logit models. This indicates that the differences among the motorcyclists’ characteristics are significant due to some unobserved factors. In addition, the impact of a motorcyclist’s BAC value being more than 0.03% becomes insignificant because of the significant difference of the standard deviation.

The motorcyclist’s BAC value being less than 0.03% is also a random parameter. It should be noted that a BAC value of less than 0.03% is not considered illegal according to Taiwanese traffic regulations. The effect of having a BAC value of less than 0.03% on motorcyclist fatality may be varied but is still significant. The random effect is also found when the motorcyclist is aged 55 years old or above. Some unobserved factors that may also affect the likelihood of motorcyclist fatality could be health conditions, reaction time, and driving experience for older motorcyclists.

Other variables with statistically insignificant standard deviations are identified as non-random parameters or have a fixed effect on motorcyclist fatality. It should be noted that the most significant coefficient estimates in the mixed logit model are larger than those in the binary logit model. This may indicate that the effects of those parameters can be found to a greater extent in the mixed logit model.

#### 5.1.2. Motorcycle vs. Motorcycle Crashes

The modeling results by both the binary logit and mixed logit models for multiple motorcycle crashes are shown in Table 3. The AIC value in the mixed logit model (8985) is less than that of the binary logit model (8991). The Chi-square (*X*^2^) statistical value (11.11) is significantly greater than the critical value (7.81) at the 95% confidence level and degree of freedom of 3. This result proves that the random parameter logit model is superior to the binary logit model for estimating the likelihood of fatal injury in multiple motorcycle crashes.

In the mixed logit model, only three parameters were identified as random parameters, including crashes occurring at midnight, side-impact collisions, and motorcyclists aged 55 years old and above. Additionally, the effects of the first two become insignificant when the random effect is considered.

The fixed effect factors of road environments include crashes occurring in the dark without streetlights, rural areas, speed limit exceeding 50 km/h, traffic island separation, and dry pavement surface. Head-on, same-direction swipe, and right-angle crashes also increase the likelihood of motorcyclist fatality. Large engine size motorcycles, male motorcyclists, and lack of a helmet also have fixed effects on the likelihood of motorcyclist fatality. Improper behaviors that increase the risk of motorcyclist fatality are only found in counterpart motorcyclists. These include speed-related violations, distracted motorcyclists, and right-of-way violations. Both subjected and counterpart motorcyclists having positive BAC values increase the risk of subjected motorcyclist fatality.

#### 5.1.3. Motorcycle vs. Vehicle Crashes

The modeling results for both binary logit and mixed logit models for the motorcycle-vehicle crashes are shown in Table 4. The AIC value in the mixed logit model (32,974) is less than that in the binary logit model (33,041). With the Chi-square (*X*^2^) statistical value and the corresponding degree of freedom, 7 (the number of random parameters), it can be found that the calculated value (80.88) is greater than the critical value (14.07) at the 95% confidence level. The null hypothesis was rejected, which once again shows that the random parameter logit model is superior to the binary logit model.

In the mixed logit model for motorcycle-vehicle crashes, seven parameters were identified as random parameters, including crashes at midnight, crashes at curve (horizontal or vertical) segments, motorcyclists aged 55 years old or above, motorcyclists not wearing a helmet and without a valid driving license, and motorcyclists with a BAC value less than 0.03% and more than 0.03% as well. Among these random parameters, the effects of curve segments and the lack of a driving license become insignificant to motorcyclist fatality due to unobserved factors.

Like with single motorcycle crashes, motorcyclists aged 55 years old or above and those with BAC values less than 0.03% also have significant random effects on motorcyclist fatality. Motorcyclists having a BAC value of more than 0.03% is also a random parameter and has a significant effect on motorcyclist fatality in motorcycle-vehicle crashes. A distinction from single motorcycle crashes is that motorcyclists not wearing a helmet is identified as a random parameter.

All other variables with a statistically insignificant standard deviation were identified as non-random parameters. Particularly, all counterpart driver’s characteristics were identified as non-random parameters. The effects of counterpart driver characteristics on the motorcyclist fatality are all fixed.

### 5.2. Risk Factors and Odds Ratios of Motorcyclist Fatality

The random parameter mixed logit model was found to be a better fitting model than the binary logit model for all three databases. The high-risk factors and their odds ratios of motorcyclist fatality based on the results of the mixed logit models are shown in Table 5. In all databases, the odds ratios were positive and greater than 1, indicating that the likelihood of motorcyclist fatality under the given parameter situation is higher than under the absence of the given parameter in the same categories. For example, in SM crashes, if the SM crash occurs at midnight, the likelihood of motorcyclist fatality is 2.17 times that of crashes occurring during any other time.

The common fatality risks of road and environment characteristics are crashes occurring at midnight, on rural roads, on roads with a speed limit of more than 50 km/h, and roads with traffic island separations. Among these common factors, crashes occurring at midnight are also a random parameter in all three models. The impact of crashes in the dark without streetlights, on curves, and on roads without lane separation on motorcyclist fatality are varied in different models.

### 5.3. Discussion

#### 5.3.1. Road and Environment Attributes

Common fatality risks of the road and environment characteristics are crashes occurring at midnight, on rural roads, on roads with a speed limit of more than 50 km/h, and on roads with traffic island separations. Among these common factors, crashes occurring at midnight are also a random parameter in all three models. This result is consistent with previous studies. Injuries resulting from after midnight night riding (0:00–07:00) generally are the most severe [15]. Motorcyclists are found to be more vulnerable during nighttime at both intersections and expressways, perhaps because of increased speeds and stronger impacts [45]. In addition, this study further confirms that the effect is more significant in SM crashes than in MM or MV crashes. A motorcyclist who crashes by him or herself at midnight (from 00 AM to 6 AM) is 2.17 times more likely to kill him or herself compared to when they have a self-crash during other periods. If a motorcycle collides with a vehicle at midnight, the likelihood of motorcyclist fatality is only 1.52 times greater than that of crashing during other periods.

A significant concern in motorcycle safety is visibility. Riding in the dark without street lighting appears to correlate with severe motorcyclist injury [7,15,46,47]. However, the lighting condition has a varied impact on the likelihood of fatality for motorcyclists in different models. Although previous research found that crashes occurring in darkness greatly increase the probability of motorcyclist fatality in both SM and MV crashes [10], the results in this study indicate that the increases are only significant in MM crashes. Theoretically, streetlights should provide better visibility for motorcyclists and vehicle drivers, decreasing the risk of crashes and severe injury. However, in SM and MV crashes, darkness (without streetlights) may not have a significantly different impact on the risk of motorcyclist fatality than other light conditions.

Motorcyclists involved in traffic crashes on rural roads tend to suffer severe or fatal injuries. This finding is also in line with previous research. Single-vehicle motorcycle crashes occurring on rural roads resulted in a higher probability of fatal injuries [11]. If motorcyclists are involved in crashes on rural roads, the probability of fatality increases by 1.14 to 1.77 times. It is believed that motorcycles or vehicles traveling on rural roads tend to have a higher speed regardless of the speed limit, leading to more severe injuries for motorcyclists.

In all motorcycle-related crashes, motorcyclists tend to have a higher fatality risk when the crashes occur on roadways with a speed limit greater than 50 kph (km/h). The likelihood of motorcyclist fatality increases by 1.67 to 2.67 times. This finding is consistent with previous studies. Roads with speed limits exceeding 50 mph have a 132% higher likelihood of a fatal injury in motorcycle-vehicle crashes [10]. Single-vehicle motorcycle crashes occurring on roads with a speed limit higher than 55 mph were more likely to result in fatal injury [11]. The effects of crashes on rural roads and roads with higher speed limits are more significant if the motorcyclist collides with a vehicle, because a higher-speed vehicle will generate a higher impact, leading to greater damage and injury.

The horizontal or vertical curvature is associated with an increase in motorcycle injury severity due to poor visibility [10]. Crashes occurring on curves increase the likelihood of motorcyclist fatality by 1.81 times. However, the increase is not significant in MM or MV crashes. In MV crashes, a crash occurring on curves is a random parameter with a significant standard deviation. The variations and unobserved parameters could be due to the curve and visibility situation not being judged and recorded precisely. No radius data is available in the crash database. Poor visibility could be due to roadside parking, street trees, or advertisement boards along the straight roads or nearby intersections. These visible obstacles are common issues in Taiwan and may not be reflected in the crash database, yet may lead to severe injury in multiple motorcycle or vehicle crashes, regardless of road curvature radios.

Lane separation facilities between motorcycles and vehicles is a common lane deployment technique in Taiwan. Lanes are separated by traffic markings or traffic islands or barriers. The initial purpose of this design was to decrease the interaction and conflict between motorcycles and four-wheel vehicles. The modeling results find that the effects of lane separations on motorcyclist fatalities are varied. In SM crashes, the separations, either by traffic markings or islands, increase the probability of motorcyclist fatality. However, the effect is not significant in MM crashes. In MV crashes, crashes occurring on roads without lane separations or roads separated by traffic islands increase the likelihood of motorcyclist fatality. Roads without lane separations indicate more conflicts between motorcycles and vehicles and may lead to more severe injury. In general, traffic islands are implemented on wide, major roads with a higher speed limit. Crashes on these roads tend to result in more severe injury in SM and MV crashes. However, due to the lower collision impact force between motorcycles, the injury severity levels are not significantly affected by different lane separations in MM crashes. Although previous research recommends that motorcycles be separated from main traffic via an exclusive motorcycle path or lane along high-speed roads [24], motorcycle lane separation types, locations, and patterns should be examined for their safety impact.

Weather conditions are not found to significantly affect the likelihood of motorcyclist fatality in all motorcycle-related crashes, but the pavement’s surface condition does affect SM and MM crashes. The probability of motorcyclist fatalities increase by 1.20 times in SM, and 1.59 times in MM crashes occurring on roads with the dry pavement surface. These findings indicate that crashes occurring on roads with wet pavement surfaces, mostly caused by rain, can generate a lower possibility of fatality in SM and MM crashes. This finding is consistent with the previous study that the wet pavement crashes are more likely to result in no injury [10].

Motorcycle riding is greatly influenced by the weather. However, past literature demonstrated some contradictory results. Weather has been reported to be a less influential factor in 98% of motorcycle accidents compared to other prevailing factors related to helmet use, age, gender, etc. [48]. Crashes occurring in fine weather increases the motorcyclist’s injury severity levels [15,46]. The wet pavement condition may force riders to travel at a low speed and therefore decrease the injury severity. However, the influence of the pavement surface condition is not significant in MV crashes.

#### 5.3.2. Crash Type and Configuration

The crash types are different in SM and MM or VM crashes. In SM crashes, a motorcycle collides with a fixed object, such as a tree, traffic device, or traffic island, leading to a 5.81 to 12.28 times probability of fatality. This result is in line with previous studies that found that collisions with stationary objects result in more severe injuries [10,16,49,50]. However, this study investigated the influence of different types of fixed objects, such as trees or utility poles, traffic devices, and traffic islands or barriers. The results show that motorcyclists hitting a tree or electrical pole leads to the highest likelihood of fatality, followed by hitting a traffic device. Roadside objects and facilities such as trees, utility poles, traffic devices, or traffic islands should be checked for appropriate placement and equipped with reflective devices or injury mitigation facilities. Buffer areas with pavement markings between the fixed object and traveling lanes are also needed.

In MM and MV crashes, as expected and consistent with previous studies [10,51], head-on and right-angle crashes result in the highest risk of fatality for motorcyclists. A motorcyclist suffers 2.84 or 1.66 times, respectively, the likelihood of fatality on average if they collide head-on or at a right-angle with another motorcycle, and 4.46 and 2.4 times with another vehicle. Head-on and right-angle collisions by vehicles result in a 566% and 227% higher likelihood of fatality [10], while no data are reported regarding head-on collisions between motorcycles.

Side impact collisions increase the likelihood of fatality for motorcyclists by 1.36 times in MV crashes. In MM crashes, the effect of side-impact collisions on motorcyclist fatality is not significant when the random parameter is considered in the model. Some unobserved factors, such as the direction of motorcyclists or the side of collision, may affect motorcyclist injury severity. Same direction or opposite direction side-impact crashes are not specified in the NPA database.

The influence of the same direction sideswipe crashes is found only in crashes between motorcycles. It increases the likelihood of motorcyclist fatality by 1.48 times on average. This finding is not noted in other studies. In addition, the impact of rear-end collisions on motorcyclist fatality is found only in MV crashes. This finding is also not consistent with previous studies; Savolainen and Mannering [10] report that rear-end collision decreases the likelihood of motorcyclist suffering incapacitating injuries by 53% and has no significant impact on motorcyclist fatality.

#### 5.3.3. Motorcycle or Motorcyclist Characteristics

Consistent with previous studies, motorcycle engines, motorcyclists’ gender, age, license, helmet use, violation behaviors, and alcohol consumption are major risk factors in motorcyclist fatality. The types and characteristics of the motorcycles have an important role in crash likelihood and severity. Past studies have also shown that motorcycle engine size is associated with motorcycle crash injury severity outcomes [7,10,15,16,21]. A large motorcycle engine size may increase the injury severity levels regardless of the control measure adopted [7,15,16,49,52,53,54]. However, this study found that the influence of motorcycle size on motorcyclist fatality is not significant in SM crashes when the unobserved heterogeneity is considered. As mentioned previously, motorcycles with engine sizes of less than 250 cc make up the greatest proportion of motorcycles. They also share the most amount of crashes. In SM crashes, other unobserved factors, such as motorcyclist attitude, riding skill, or riding gear used, may be essential factors for the injury severity in the crashes. Motorcycles with engine sizes of more than 250 cc are allowed to travel on expressways and tend to travel at a higher speed; thus, a motorcyclist riding a motorcycle with an engine size of more than 250 cc is 3.27 and 2.58 times more likely to sustain fatal injuries in MM and MV crashes.

Motorcyclists riding a large engine-sized motorcycle (more than 250 cc) tend to have fatal injuries or lead the counterpart motorcyclists dying in motorcycle-involved crashes. That could be because the motorcyclists who ride a large engine size motorcycle tend to have higher speeds which results in severe injury either in single motorcycle or multiple vehicle crashes. This result is consistent with most previous studies [4,9,16,21].

Regarding motorcyclist gender, male motorcyclists not only share the major portion (around 60%) of crashes, but also have a higher risk of fatality than female motorcyclists in Taiwan. The likelihood of male motorcyclist fatality is 1.87 times that of female motorcyclist in SM crashes, 1.7 times in MM crashes and 1.57 times in MV crashes. This finding is in line with previous studies that found female motorcyclists are 20% more likely to be involved in crashes with incapacitating injuries [10]. Several studies confirmed that male motorcyclists tend to have riskier behaviors associated with increased risks of crashes and disregard traffic regulations and motorcycle safety checks [45,49,53,55,56,57].

Although young motorcyclists also tend to have risky behaviors and ignore traffic regulations, older motorcyclists are more likely to be involved in severe injury crashes [10,14,15], particularly those aged 55 years old or above. Youth motorcyclists are 1.38 times more likely to be killed in SM crashes than middle-aged motorcyclists. However, the fatal risk of youth motorcyclists is not significant in MM and MV crashes. Motorcyclists aged 55 and above are 1.86 times more likely to be killed in SM crashes. Elderly motorcyclists is a random parameter, indicating that some unobserved parameters related to older motorcyclists may also affect the injury severity in the SM crashes, such as motorcyclists’ health condition, riding skills, or experience. The likelihood of fatality for older motorcyclists increases 3.89 times in MM crashes and 2.83 times in MV crashes.

Old road users are a major traffic safety issue currently. They tend to suffer severe injury in traffic crashes because they have a slower reaction time and lack sensory and perceptual abilities and physical resiliency to motorcycle crashes when compared to younger road users [10,14,15]. Periodic driving license re-examination or health condition checks may be desired, especially for older motorcyclists. In Taiwan, no check or license renewal is required until 75 years old if the license has not been suspended.

Helmets have been confirmed to be effective equipment in decreasing motorcyclist injury severities in motorcycle crashes in several studies [10,17,18,19]. This study also found that motorcyclists not wearing a helmet are more likely to sustain a fatal injury by 5.58 times in SM crashes, 7.16 times in MM crashes, and 2.95 times in MV crashes. Even though wearing a helmet is mandatory for riding a motorcycle in Taiwan, the requirement is still often violated, which increases risk of death. Greater enforcement is needed to increase the usage of helmets.

Holding a valid license is also a basic and essential requirement for riding a motorcycle safely. Previous studies found that motorcyclists without a valid driving license have a higher probability of fatality than those with a valid license [20,57]. However, this is only in SM crashes. In MV crashes, lacking a valid riding license is a random parameter, indicating that some unobserved factors may affect the riding skills or risk recognition. Although motorcyclist education and licensing play key roles in reducing motorcycle crashes and injuries, little is known about what constitutes effective rider training and licensing [58]. However, driver education programs are proven to produce safer drivers, especially for young or beginner drivers [59]. Teens taking driver education are less likely to be involved in crashes or to receive a traffic violation during their first two years of driving [60]. In Taiwan, education or training is not required before taking the motorcycle (less than 250 cc) licensing exam. Drivers only need to pass a traffic regulation examination and a closed field riding test to receive a valid riding license. Thus, those who have a valid riding license may not be capable of risk recognition and crash prevention, especially when facing conflicts with other motorcycles and vehicles.

Improper riding behaviors or violations also increase the likelihood of fatality. The effects vary among different types of behaviors. Distracted-driver and speed-related violations significantly increase the probability of motorcyclist fatality by 1.24 and 2.46 times in SM crashes. In MV crashes, speed-related violations, right-of-way violations, and improper turns are critical factors that increase the likelihood of motorcyclist fatality. However, motorcyclist violations are only found to affect counterpart motorcyclists in MM crashes. The subjected motorcyclist’s violations do not significantly affect the likelihood of their fatality in MM crashes. This indicates that the threat of fatal injury comes from counterpart motorcyclists rather than themselves in MM crashes. In MV crashes, the fatality threat is not only from the counterpart vehicles but also from themselves. Improper turns result in the highest likelihood of motorcyclist fatality (2.45 times). The motorcycle lane design that forces most motorcycles to travel along roadside lanes may lead some motorcycles to make a left turn from right-side lanes, generating conflicts with vehicles traveling on the left-side or middle lanes.

Alcohol consumption increases the likelihood of fatality for motorcyclists. Motorcyclists whose BAC values are positive have a higher probability of fatality, regardless of if the BAC value exceeds the legally limited value. In SM crashes, motorcyclists having a positive BAC value of less than 0.03% increases the likelihood of fatality by 3.02 times, while the increase is not significant for those with a BAC value of more than 0.03%. A motorcyclist’s BAC value is found as a random parameter, indicating that some unobserved factors may affect the BAC test and the influence of the BAC values on injury severities.

Motorcyclists who have a positive BAC value of less than 0.03% have an increased probability of fatality by 6.39 times in MM crashes, and by 3.72 times in MV crashes. Disregarding those with an unknown BAC value, motorcyclists whose BAC is positive, but less than 0.03% g/dL have the highest probability of fatality. This indicates that even though the motorcyclist’s BAC value does not exceed the legally limited value, the influence of alcohol still increases the risk of motorcyclist fatality. Driving or riding should be prohibited if drivers or motorcyclists have consumed any amount of alcohol and have a positive BAC value.

#### 5.3.4. Counterpart Motorcyclist and Driver Characteristics

The characteristics of counterpart motorcyclists or drivers are barely discussed in literature beyond vehicle type and engine size. Motorcycles that have crashes with greater engine-size motorcycles or large vehicles have an increased likelihood of motorcyclist fatality. A motorcyclist is 4.26 times more likely to be killed when they collide with a greater engine size motorcycle and is 8.79 times more likely to suffer fatal injury when they collide with a large truck or bus. This is in line with previous studies that found collisions with a bus/coach increase motorcyclist injury severity [15,21,46]. Collisions with heavier vehicles result in more severe injuries [15,16,49,50].

Counterpart motorcyclists and vehicle drivers without a valid driving license increase the likelihood of the subjected motorcyclist’s fatality by 1.82 times in MM crashes, and 1.76 times in MV crashes. Unlicensed motorcyclists or drivers are critical risk factors for motorcyclist fatality–not only for themselves in SM crashes, but also for counterpart motorcyclists. Improper behaviors or violations by counterpart motorcyclists or drivers also increase the likelihood of the subjected motorcyclist’s fatality. A motorcyclist colliding with a motorcycle with a speed-related and right-of-way violation is 2.22 and 1.45 times, respectively, more likely to be killed. Similarly, a motorcyclist who collides with a vehicle with a speed-related and right-of-way violation is 1.69 and 1.65 times, respectively, more likely to be killed. Distracted motorcyclists also increase the likelihood of counterpart fatality by 1.82 times, while distracted vehicle drivers are not found to have a significant impact on fatality.

Motorcyclists or vehicle drivers with a positive BAC value also threaten a counterpart motorcyclist’s life in crashes. Motorcyclists with a positive BAC increase the probability of counterpart fatality by 2.13 to 2.7 times. Motorcyclists who collide with a vehicle driver with a positive BAC are 1.76 to 3 times more likely to be killed. The likelihood of fatality increases as the driver’s BAC value increases.

### 5.4. Novel Findings

Motorcycle safety is a worldwide road safety issue. Numerous studies around the world have identified several critical factors that affect the likelihood of motorcyclist fatality. Most factors contributing to motorcyclist fatality found in previous studies were also found in this study, such as older motorcyclists, not wearing a helmet, and speeding violations. These are common factors of motorcyclist fatality due to physiological weakness, reduced protection, or strong impact. However, considering the heterogeneity across potential factors and the road, traffic components, and lane usage management attributes in Taiwan, this study found several novelties affecting motorcyclist fatalities.
This study analyzed the crashes between motorcycles separately, which was not performed in previous studies. Novelty results in MM crashes include crashes in the dark without streetlights and same direction swipe crashes increasing the likelihood of motorcyclist fatality. Motorcyclists without a valid driving license do not affect the likelihood of their fatality but do impact the counterpart motorcyclists’ lives in MM crashes.Random parameters which consider the heterogeneity were found in all models–for example, crashes occurring at midnight and elderly motorcyclists in SM and MM crashes. However, with significantly large standard deviations, some become less impactful on the likelihood of motorcyclist fatality, such as engine size in SM crashes.Crashes occurring on roads with a traffic island lane separations increase the likelihood of motorcyclist fatality in all crashes. The increase in fatality is only significant on roads with pavement marking lane separations in SM crashes.The fatality risk of youth and unlicensed motorcyclists is only significant in SM crashes. This indicates that license status is not a critical factor in motorcyclist fatality in crashes with other motorcycles or vehicles. Even though they have a valid license, they may still lack riding skills or risk recognition of interacting with other motor vehicles.Alcohol’s effect on motorcyclist fatality was analyzed by considering the legal limited BAC for riding or driving. Motorcyclists with positive BAC values less than the legal limited value have a higher likelihood of fatality in all crashes.Counterpart motorcyclist violation and behaviors, such as speeding, driving distracted, right of way violations, and not having a valid license have a greater impact on motorcyclist fatality than violations by subjected motorcyclists themselves.

### 5.5. Research Limitations

This study was based on police-reported crash data. The results are limited within the variables or parameters recorded by the police. Some potential factors or parameters, such as motorcyclist health conditions, mental or physical issues, may not be reflected in the results. For example, the mixed logit modeling results found several random parameters that may be affected by unobserved parameters. However, the identification of unobserved parameters requires further study.

The fatality rates are less than 1% in several situations, especially in MM crashes. The imbalanced data issues were not considered in this study. Due to the large sample size in each crash database (more than 120 thousand motorcyclists), the high number of fatalities is not a rare event. The effect of rare events on logit modeling may not be significant. However, further study address the imbalance of data in predicting the likelihood of motorcyclist fatality is needed.

The motorcycle population, engine size, traffic regulation, road alignment, and environment are varied among different countries and cities. Some factors that affect the likelihood of motorcyclist fatality may not be consistent in different research areas. This study identified factors affecting motorcyclist fatality under a particular road traffic environment, such as numerous computing motorcycles with engine sizes of less than 250 cc, and the restrictions on riding space and regulations. Some results of this study may not apply to other cities or countries. For example, crashes occurring on roads with lanes separated by traffic islands had an increased likelihood of motorcyclist fatalities in this study. However, this finding provides an opposite opinion for a research statement that recommends motorcycles be separated from the main traffic by implementing an exclusive motorcycle path [24].

A recent study found that the use of individual transportation modes, including motorcycles, increased during the COVID-19 pandemic [61]. This may also increase the likelihood of motorcycle-involved crashes or injury severity. However, the impact of the COVID-19 pandemic on Taiwanese society was only for a short period, from March to May in 2020. The effect of the COVID-19 pandemic on motorcyclist fatality is limited and was not considered in the modeling process. A study which focuses on the long-term observations on the impact of the COVID-19 pandemic on road safety and fatal injuries is needed.

## 6. Conclusions and Recommendations

### 6.1. Conclusions

To better understand the factors contributing to motorcyclist fatality, traditional binary logit and random parameter mixed logit models were applied to identify the critical factors affecting the likelihood of motorcyclist fatality based on three separated types of motorcycle-involved crashes. The results revealed that the mixed logit models had better fitting performances in the three separated models. Common factors about road attributes, consistent with previous studies, include crashes occurring at midnight, rural areas, higher speed limits, and lane separation by traffic island. The effect of curves on the likelihood of motorcyclist fatality was only found in SM and MV crashes. The impact of roads without lane separations on motorcyclist fatality was only significant in MV crashes due to the limited space between motorcycles and vehicles.

In line with previous studies, collisions with fixed objects and head-on collisions increase the probability of motorcyclist fatality. This study further found that collisions with trees or utility poles result in the highest probability of fatal injuries in SM crashes. Same direction swipe crashes also increase the likelihood of motorcyclist fatality in MM crashes.

Regarding motorcyclist characteristics, common factors (consistent with most studies) include being of the male sex, elderly motorcyclists, and not wearing a helmet. These factors increase the risk of fatality in all crashes. Partially inconsistent with previous studies, the impact of large-engine-size motorcycles on motorcyclist fatality was only found to be significant in MM and MV crashes. In SM crashes, the effect is insignificant if heterogeneity is considered in the modeling process. Young motorcyclists have a higher risk of fatality in only SM crashes, due to the lack of riding skills. The effect of alcohol is most significant when the motorcyclist’s BAC value is positive but less than the legal limited value.

Lack of a valid riding license is a critical factor of fatality for motorcyclists themselves in SM crashes but not in MM and MV crashes. However, it is a critical factor of fatality for counterpart motorcyclists in MM and MV crashes. In addition, motorcyclists’ improper behaviors or violations are not significant factors increasing their likelihood of fatality in MM crashes but are risk factors of fatality in MV crashes. Motorcyclists tend to be killed if they collide with large-engine-size motorcycles and vehicles, unlicensed motorcyclists, or vehicle drivers with traffic violations, such as speeding, violating the right of way, or with a positive BAC value.

### 6.2. Recommendations

According to the findings, young and unlicensed motorcyclists have a high risk of fatality in SM crashes, while unlicensed motorcyclists do not represent a critical factor in MM and MV crashes. This indicates that good training and education is needed to obtain a motorcycle license to enhance youth motorcyclists’ riding skills and risk recognition abilities. Having periodic health checks and license renewal requirements is also recommended, especially for older motorcyclists, to ensure that motorcyclists are still capable of properly riding a motorcycle.

Alcohol is a critical factor affecting motorcyclists and counterpart fatalities. Motorcyclists with a positive BAC value within the legal limited value have a higher likelihood of fatality in all crashes. It is recommended that driving or riding should be prohibited if the drivers or motorcyclists consume any amount of alcohol or have a positive BAC value.

Roads with motorcycle lanes separated by traffic islands increase the risk of motorcyclist fatality. Motorcycle lane separation types, locations and patterns should be examined for their demand and safety impact. Roadside objects and facilities such as utility poles, traffic devices, and traffic islands should be checked for their appropriate locations and equipped with reflective devices or injury protection facilities.

Law enforcement should focus on motorcyclists and drivers who tend to increase the probability of fatalities, such as those that are unlicensed, not wearing a helmet, speeding, and those who violate the right of way and have improper turns.

## Figures and Tables

**Table 1 ijerph-19-08411-t001:** Crash and fatality distribution for SM, MM, and MV crashes.

Variable	Single Motorcycle Crashes			Motorcycle vs. Motorcycle Crashes			Motorcycle vs. Vehicle Crashes		
	No. of Motorcyclist	% of Motorcyclist	% of Fatality	No. of Motorcyclist	% of Motorcyclist	% of Fatality	No. of Motorcyclist	% of Motorcyclist	% of Fatality
Total	122,710	100%	1.29%	771,029	100%	0.08%	735,270	100%	0.46%
Fatality	1586	1.3%	100%	628	0.1%	100%	3400	0.5%	100%
*Road environment attributes*									
Midnight (00–06)	10,201	8.3%	3.61%	16,691	2.2%	0.21%	22,741	3.1%	1.30%
Dark without streetlight	3686	3.0%	1.98%	4400	0.6%	0.25%	5105	0.7%	1.04%
Rural	43,911	35.8%	1.96%	126,735	16.4%	0.17%	165,834	22.6%	0.90%
Speed limit > 50 km/h	10,032	8.2%	2.48%	28,296	3.7%	0.27%	48,704	6.6%	1.29%
Intersection	34,277	27.9%	0.55%	516,177	66.9%	0.08%	482,033	65.6%	0.46%
Horizontal or vertical curve	16,768	13.7%	2.44%	15,244	2.0%	0.15%	17,357	2.4%	0.97%
Straight segment	68,848	56.1%	1.39%	229,700	29.8%	0.08%	225,465	30.7%	0.44%
Lane Separation-marking	56,083	45.7%	1.36%	326,924	42.4%	0.09%	308,850	42.0%	0.39%
Lane separation-traffic Island	29,736	24.2%	1.41%	123,083	16.0%	0.09%	138,479	18.8%	0.50%
Pavement surface-dry	98,440	80.2%	1.42%	698,125	90.5%	0.09%	636,457	86.6%	0.48%
Pavement surface-wet	24,270	19.8%	0.78%	72,904	9.5%	0.05%	98,813	13.4%	0.35%
Inclement weather	17,706	14.4%	0.78%	57,624	7.5%	0.04%	81,115	11.0%	0.32%
*Crash characteristics/types*									
Fall over	80,518	65.6%	0.47%						
Run off	4363	3.6%	4.42%						
Hit tree/pole	5301	4.3%	7.68%						
Hit traffic device	1150	0.9%	6.52%						
Hit island/barrier	8127	6.6%	4.41%						
Swipe crash-same direction				92,471	12.0%	0.09%	104,399	14.2%	0.32%
Swipe crash-opposite direction				29,184	3.8%	0.09%	27,850	3.8%	0.36%
Head-on crash				10,713	1.4%	0.22%	6716	0.9%	2.14%
Read-end crash				107,698	14.0%	0.05%	64,028	8.7%	0.47%
Side-impact crash				280,617	36.4%	0.08%	299,656	40.8%	0.40%
Right-angle crash				111,152	14.4%	0.12%	81,471	11.1%	1.01%
*Motorcycle/list characteristics*									
LHM (250 cc and above)	1905	1.6%	3.83%	5470	0.7%	0.26%	7245	1.0%	1.16%
Male	76,972	62.7%	1.69%	446,243	57.9%	0.10%	423,164	57.6%	0.55%
Young (24 years old and less)	46,031	37.5%	0.91%	242,881	31.5%	0.04%	253,458	34.5%	0.26%
Middle Age (25–54years old)	51,279	41.8%	1.31%	364,158	47.2%	0.05%	322,145	43.8%	0.33%
Aged (55 years old and above)	25,400	20.7%	1.97%	163,990	21.3%	0.23%	159,667	21.7%	1.04%
without wearing a helmet	1309	1.1%	11.77%	5833	0.8%	1.05%	6927	0.9%	3.44%
without valid driving license	13,967	11.4%	3.19%	71,661	9.3%	0.16%	78,478	10.7%	1.04%
Districted	41,481	33.8%	1.40%	160,331	20.8%	0.07%	235,647	32.0%	0.35%
Speed violation	7446	6.1%	2.10%	44,411	5.8%	0.06%	50,013	6.8%	0.48%
Improper turn				51,739	6.7%	0.12%	21,135	2.9%	0.89%
Right of way violation				155,393	20.2%	0.12%	97,799	13.3%	0.96%
Impaired	11,335	9.2%	2.87%	2792	0.4%	0.65%	6029	0.8%	2.22%
Other violations	47,376	38.6%	1.04%	204,256	26.5%	0.08%	158,307	21.5%	0.37%
Motorcyclist’s BAC (<0.3%)	926	0.8%	7.88%	2812	0.4%	0.71%	3378	0.5%	4.50%
Motorcyclist’s BAC (≥0.3%)	10,378	8.5%	3.38%	6848	0.9%	0.56%	9848	1.3%	2.50%
Motorcyclist’s BAC unknown	3332	2.7%	14.41%	11,299	1.5%	0.60%	6698	0.9%	12.45%
*Counterpart motorcycle/vehicle characteristics*									
LHM (250 cc+)				5469	0.7%	0.51%			
Large Car (Bus & Truck)							22,659	3.1%	3.37%
Without driving license				71,676	9.3%	0.16%	26,380	3.6%	0.85%
Districted				160,293	20.8%	0.12%	91,828	12.5%	0.77%
Speed related violations				44,408	5.8%	0.16%	26,302	3.6%	1.43%
Improper turns				51,750	6.7%	0.06%	107,954	14.7%	0.21%
Right of way violation				155,393	20.2%	0.12%	107,954	14.7%	0.21%
Impaired				2791	0.4%	0.29%	2076	0.3%	1.69%
Other violations				204,329	26.5%	0.06%	195,570	26.6%	0.36%
Motorcyclist’s driver’s BAC (+<0.3%)				2811	0.4%	0.39%	3788	0.5%	1.32%
Motorcyclist’s driver’s BAC (≥0.3%)				6847	0.9%	0.28%	5633	0.8%	1.76%
Motorcyclist/driver’s BAC unknown				11,567	1.5%	0.14%	20,657	2.8%	0.61%

**Table 2 ijerph-19-08411-t002:** Binary logit and heteroscedastic logit model results for single motorcycle crashes.

Variable	Binary Logit Model		Mixed Logit Model	
	Coefficient Estimate	*p*-Value	Coefficient Estimate (Standard Deviation)	*p*-Value
Intercept	−7.453	0.0000	−7.620	<0.001
*Crash and environment characteristics*				
Midnight (00–06)	0.742	<0.001	0.774	<0.001
Rural	0.139	0.0191	0.130	0.0395
Speed limit > 50 km/h	0.479	<0.001	0.514	<0.001
Horizontal or vertical curve	0.554	<0.001	0.591	<0.001
Lane separation-markings	0.251	<0.001	0.266	<0.001
Lane separation-traffic Island	0.473	<0.001	0.489	<0.001
Pavement surface-dry	0.188	0.0249	0.183	0.0368
*Crash type*				
Run off	1.630	<0.001	1.698	<0.001
Hit tree/pole	2.389	<0.001	2.508	<0.001
Hit traffic device	2.156	<0.001	1.897 (1.106)	<0.001 (0.0392)
Hit island/barrier	1.684	<0.001	1.759	<0.001
*Motorcyclist characteristics*				
Male	0.607	<0.001	0.624	<0.001
Young (<25 years old)	0.310	<0.001	0.321	<0.001
Aged (55 years old+)	0.877	<0.001	0.623 (0.902)	<0.001 (<0.001)
Without wearing a helmet	1.534	<0.001	1.719	<0.001
Without a valid driving license	0.445	<0.001	0.487	<0.001
Distracted	0.214	<0.001	0.220	<0.001
Speed violation	0.868	<0.001	0.900	<0.001
LHM (250 cc+)	0.732	<0.001	0.019 (1.554)	0.9695 (n.s.) (0.0021)
Motorcyclist’s BAC (+<0.3%)	1.827	<0.001	1.105 (1.696)	0.0148 (<0.001)
Motorcyclist’s BAC (≥0.3%)	0.828	<0.001	0.288 (1.228)	0.2640 (n.s.) (<0.001)
Motorcyclist’s BAC unknown	3.152	<0.001	3.252	0.0392
*Model statistics*				
*LL0*	−8472.6		−8472.6	
*LL*	−6052.7		−6041.1	
*R^2^*	0.286		0.287	
*AIC*	12,151		12,138	

Note: n.s.: not significant at the 95% confidence interval.

**Table 3 ijerph-19-08411-t003:** Binary logit and heteroscedastic logit model results for crashes between motorcycles.

Variable	Binary Logit Model		Mixed Logit Model	
	Coefficient Estimate	*p*-Value	Coefficient Estimate (Standard Deviation)	*p*-Value
Intercept	0.209	<0.001	−9.705	<0.001
*Crash and environment characteristics*				
Midnight (00–06)	0.913	<0.001	−0.526 (1.795)	0.3939 (n.s.) (<0.001)
Dark without streetlight	0.820	0.0083	0.839	0.0080
Rural	0.552	<0.001	0.561	<0.001
Speed limit > 50 km/h	0.872	<0.001	0.885	<0.001
Separation-Traffic Island	0.249	0.0257	0.256	0.0235
Pavement surface-dry	0.456	0.0101	0.464	0.0094
*Crash type*				
Head-on collision	1.032	<0.001	1.035	<0.001
Swipe collision—same direction	0.393	0.0042	0.397	0.0041
Right angle collision	0.496	<0.001	0.501	<0.001
Side impact collision	0.275	0.0096	−0.168 (0.985)	0.4697 (n.s.) (<0.001)
*Motorcyclist characteristics*				
Male	0.524	<0.001	0.528	<0.001
Aged (55 years old+)	1.573	<0.001	1.358 (0.688)	<0.001 (0.011)
Without wearing helmet	1.895	<0.001	1.946	<0.001
LHM (250 cc+)	1.162	<0.001	1.181	<0.001
Motorcyclist’s BAC (+<0.3%)	1.823	<0.001	1.853	<0.001
Motorcyclist’s BAC (≥0.3%)	1.650	<0.001	1.686	<0.001
Motorcyclist’s BAC unknown	1.952	<0.001	1.995	<0.001
*Counterpart motorcyclist characteristics*				
LHM (250 cc+)	1.422	<0.001	1.448	<0.001
Without valid driving license	0.588	<0.001	0.597	<0.001
Speed related violation	0.788	<0.001	0.794	<0.001
Distracted	0.464	<0.001	0.466	<0.001
Right of way violation	0.376	<0.001	0.376	<0.001
Motorcyclist’s BAC (+<0.3%)	0.994	<0.001	1.022	<0.001
Motorcyclist’s BAC (≥0.3%)	0.758	<0.001	0.774	<0.001
*Model statistics*				
*LL0*			−5095	
*LL*	−4470.251		−4465	
*R^2^*	0.123		0.124	
*AIC*	8991		8985	

Note: n.s.: not significant at the 95% confidence interval.

**Table 4 ijerph-19-08411-t004:** Binary logit and Mixed logit model results for crashes between motorcycle and vehicle.

Variable	Binary Logit Model		Mixed Logit Model	
	Coefficient Estimate	*p*-Value	Coefficient Estimate (Standard Deviation)	*p*-Value
Intercept	0.070	<0.001	0.266	<0.001
*Crash and environment characteristics*				
Midnight (00–06)	0.715	<0.001	0.419 (0.927)	0.034 (<0.001)
Rural	0.553	<0.001	0.572	<0.001
Speed limit > 50 km/h	0.926	<0.001	0.981	<0.001
Horizontal or vertical curve	0.452	<0.001	0.051 (1.087)	0.834 (n.s.) (<0.001)
No separation	0.138	<0.001	0.147	<0.001
Separation-traffic Island	0.168	<0.001	0.170	0.0021
*Crash type*				
Head-on collision	1.373	<0.001	1.496	<0.001
Right angle collision	0.822	<0.001	0.875	<0.001
Side impact collision	0.290	<0.001	0.308	<0.001
Rear-end collision	0.576	<0.001	0.594	<0.001
*Motorcyclist characteristics*				
Male	0.423	<0.001	0.453	<0.001
Middle age (25–54 years old)	0.204	<0.001	0.200	<0.001
Aged (55 years old+)	1.178	<0.001	1.042 (0.651)	<0.001 (0.005)
Without wearing helmet	1.297	<0.001	1.081 (0.986)	<0.001 (<0.001)
Without valid driving license	0.325	<0.001	−0.165 (1.148)	0.1799 (n.s.) (<0.001)
BHM (250 cc+)	0.938	<0.001	0.948	<0.001
Speed violation	0.199	0.0089	0.216	0.0061
Right of way violation	0.590	<0.001	0.615	<0.001
Improper turn	0.856	<0.001	0.894	<0.001
Impaired	0.252	0.0382	0.260	0.065 (n.s.)
Motorcyclist’s BAC (+<0.3%)	2.176	<0.001	1.315 (1.739)	<0.001 (<0.001)
Motorcyclist’s BAC (≥0.3%)	1.499	<0.001	0.837 (1.424)	0.006 (<0.001)
Motorcyclist’s BAC unknown	3.644	<0.001	3.824	<0.001
*Counterpart driver characteristics*				
Large vehicle (truck or bus)	2.049	<0.001	2.174	<0.001
Without valid driving license	0.538	<0.001	0.564	<0.001
Speed violation	0.644	<0.001	0.674	<0.001
Right of way violation	0.488	<0.001	0.501	<0.001
Motorcyclist’s BAC (+<0.3%)	0.511	<0.001	0.567	<0.001
Motorcyclist’s BAC (≥0.3%)	1.032	<0.001	1.097	<0.001
*Model statistics*				
*Number of observations*				
*LL0*	−21,672		−21,672	
*LL*	−16,491		−16,450	
*R^2^*	0.239		0.241	
*AIC*	33,041		32,974	

Note: n.s.: not significant at the 95% confidence interval.

**Table 5 ijerph-19-08411-t005:** Odds ratios of motorcyclist fatality based on mixed logit models.

Variable	SM Crashes	MM Crashes	MV Crashes
*Road and environment characteristics*			
Midnight (00–06)	2.17 (*r*)	0.59 (*r*; n.s.)	1.52 (r)
Dark without streetlight	-	2.31	-
Rural	1.14	1.76	1.77
Speed limit > 50 km/h	1.67	2.44	2.67
Horizontal or vertical curve	1.81		1.05 (*r*; n.s.)
Lane separation-No separation	-		1.16
Lane separation-markings	1.30		-
Lane separation-Traffic Island	1.63	1.29	1.19
Pavement surface-dry	1.20	1.59	-
*Crash type*			
Run off	5.46	-	-
Hit tree/pole	12.28	-	-
Hit traffic device	6.67	-	-
Hit island/barrier	5.81	-	-
Swipe crash-same direction	-	1.49	
Head-on collision	-	2.84	4.46
Right angle collision	-	1.66	2.40
Side impact collision	-	0.85 (*r*; n.s.)	1.36
Rear-end collision	-		1.81
*Motorcycle and Motorcyclist characteristics*			
BHM (250 cc+)	1.02 (*r*; n.s.)	3.27	2.58
Male	1.87	1.70	1.57
Young (<25 years old)	1.38	-	-
Middle age (25–54 years old)	-	-	1.22
Aged (55 years old+)	1.86 (*r*)	3.89 (*r*)	2.83
Without wearing helmet	5.58	7.16	2.95
Without valid driving license	1.63		0.85 (*r*; n.s.)
Distracted	1.25		-
Speed violation	2.46		1.24
Right of way violation	-		1.85
Improper turn	-		2.45
Motorcyclist’s BAC (+<0.3%)	3.02	6.39	3.72
Motorcyclist’s BAC (≥0.3%)	1.33 (*r*; n.s.)	2.57 (*r*; n.s.)	2.31
Motorcyclist’s BAC unknown	25.84	7.05	45.81
*Counterpart motorcyclist/driver characteristics*			
LHM (250 cc+)		4.26	-
Large vehicle (truck or bus)		-	8.79
Without a valid driving license		1.82	1.76
Distracted		1.59	-
Speed related violations		2.22	1.96
Right of way violations		1.45	1.65
Motorcyclist’s BAC (+<0.3%)		2.72	1.76
Motorcyclist’s BAC (≥0.3%)		2.21	3.00

Note: r: random parameter. n.s.: not significant at the 95% confidence interval.

## Data Availability

The data presented in this study are available on request from the author.

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
