# Peer review of "Investigating the Difference in Factors Contributing to the Likelihood of Motorcyclist Fatalities in Single Motorcycle and Multiple Vehicle Crashes"

_ijerph, 2022, doi:10.3390/ijerph19148411_

Round 1
Reviewer 1 Report
The paper is well written, and I recommend to accept the paper with some minor revision. My major comments are below:
(1) there have been some related works to identify key factors form massive data, please charify the advantages of the proposed method. You can refer to doi.org/10.1016/j.eswa.2021.114902.
(2) Please delte the word of "background" in the abstract.
(3) I suggest the authors to remove the tables in appendixs.
(4) I wonder whether the conclusion can be used in other cities, please provide some illustration for the robustness analysis of the proposed method.
Author Response
Please check the attached file for all responses, Thanks

Reviewer 2 Report
The present study focuses on modeling the factors contributing to the motorcyclist fatality in motorcycle-related crashes in Taiwan. The research topic is important.
Extensive language problems must be fixed. Use just present simple tense towards all text. Please, use a more connective way while presenting section 2 (literature review). Use large tables such as table 1 in section 3 for the appendix to increase traceability. The methodology section is poor. Be more elaborative in terms of methodologic reasoning and be more specific in terms of the applied method, many logit models could be applied here, also the functioning of R squared, multicollinearity, and AIC in the model analysis should be mentioned here. The results section is not traceable, please reorganize, as well as I expect a better level of interpretation of the result. I suggest focusing on the fundamental gain of this study in the conclusion, that law enforcement suggestions should be supported by the increase in motorcycle usage during the COVID19 pandemic from the literature (e.g., https://doi.org/10.1080/19427867.2021.1901011). Lastly, I suggest more principal keywords for this article, and please connect background and method: to each other in the abstract.
Author Response
please check the attached file for all responses, Thanks

Reviewer 3 Report
The study is incomplete.
Separating from single motorcycle crashes to multiple-vehicle crashes is a good idea to explore deeper into various contributing factors to motorcyclist fatality risk. However, this does not provide anything new to the readers. Especially when people are talking a lot about individual-level unobserved heterogeneity, and this becomes a must-to-solve issue in recent logit models. For example, the random parameter models.
Other minor issues are:
1) two decimal digits are reported in Table 1, I do not think it is necessary;
2) missing unit for speed limit
3) Driver/Motorcyclist’s cause of crash, not sure how to determine who is at-fault;
4) please provide supporting evidence for groupings of BAC
5) definition of *
6) crash attributes, road environment attributes, road environment attributes, road environment attributes should stay in one model to capture their mutual effect
7) missing author contributions, and other info. required by the journal.
Author Response

(The authors gave the same response as above.)

Reviewer 4 Report
In the study, author conducted an analysis of the crash database in order to investigate and determine factors that contribute to the likelihood of motorcyclists fatalities in Taiwan. The database consisted five years of crashes (single-motorcycle, motorcycle-motorcycle, and motorcycle-vehicle crashes) and author used logistic regression to analyze the data. Although I appreciate author's effort in conducting an analysis on such a number of variables, I'm missing the novelty of the work and proper discussion of the results. Overall, the manuscript in the current form, does not provide anything new to the existing literature and in fact majority of the results, i.e. influential factors are already well known and understood in the literature. More detailed comments are presented below:
- Introduction – In general, the Introduction lacks of a more „story telling“ approach and in some cases sentences are „chopped“ and unconnected. For example, „In 2020, about 61% (1,837 out of 3,000) 32 of all traffic fatalities were motorcyclists. Among all motorcycle-related fatal crashes, 33 about 28% (520 out of 1,837) were single-motorcycle crashes. The number of motorcyclists 34 who died in single-motorcycle crashes increased from 451 in 2017 to 520 in 2020. Fatal 35 single-motorcycle (SM) crashes have become a major issue in motorcycle-related crashes.“, in my view author jumps from one sentence to another without properly connecting them and writing a story which can be more easily followed.
- I would suggest that author merges Introduction section with section Related Works and creates more easy-to-follow introduction where he will elaborate on the previous findings and factors which have been analyzed by other researchers.
- Besides aforementioned, after the presentation of the problem and literature findings, author should clearly state what are the gaps in literature and describe what is main aim of the study as well as how this study will contribute to the existing body of literature.
- Row 95: „…NPA (National Police Administration)“. Please write the full name and abbreviation in the bracket.
- Row 132: „A total of 122,710, 771,029, and 735,270 motorcyclists involved in SM, MM, and MV crashes, respectively, were considered in the final dataset for analysis.“ – could author rephrase this sentence and separate the number so it is easier to read them?
- Table 1 is too long and therefore hard to read so I suggest that author moves it to the Appendix. In such way, reader can, if interested, go a check the table and the overall manuscript would be much easier to read.
- I would suggest to author to not to copy-paste the tables from Excel, or at least improve the layout of the tables.
- In general, although I appreciate the author's work, I have a concern about some of the conclusions from the analysis due to the differences in sample size. For example in row 281 it is written „For motorcyclist characteristics, male motorcyclists have a higher risk of fatality than female motorcyclists in SM, MM, and MV crashes (as shown in Table 5).“. However, if we look at the Table 1, we can see that there is more than 60% of male drivers compared to 30% of female drivers (there are more of such examples). How do author justify their conclusions with such a differences in sample sizes? Author should clearly state this when discussing the results and note that this is one of the limitations of the study.
- In the Discussion section author mostly repeats main results described earlier. In only few instances obtained results are compared with available literature and properly discussed. Author should rewrite the Discussion part in a way that he connects his results with the literature and elaborates more on the possible reasons for such results. Moreover, instead of just analyzing each factor by itself, I would be more interested in connecting the factors and discussing a more broader causes of trend sin motorcyclists crashes. Current body of literature clearly highlight interconnection of different factors. In addition, paragraph addressing the limitations of the study (and there are several of them – from uneven samples to potential inaccuracies of police reports etc.).
- Conclusion: „To better understand the factors contributing to motorcyclist fatality, five years (2016-2020) of police-reported motorcycle-involved crashes on non-freeways and non-expressways were separated into three datasets of single-motorcycle, motorcycle-motorcycle, and motorcycle-vehicle crashes. Four logistic regression models based on these three datasets were conducted to identify the factors contributing to motorcyclist fatality and to examine the relevant variables for determining the odds of motorcyclist fatality.“ – Author does not need to repeat what he already describe the methodology etc., and in my view such sentences are unnecessary so I suggest deleting them or at least rephrasing and shortening them.
- Besides law enforcement, don't author think that some educational programs may also play important role? More specifically, if such programs are properly designed and focus to specific group they could provide some benefits, as confirmed in studies such as: https://www.springerprofessional.de/en/the-effect-of-road-safety-education-on-the-relationship-between-/16546182 ; Mayhew, D., et al. (2017). Evaluation of beginner driver education in Oregon. Safety 3, (1), 9. https://doi.org/10.3390/safety3010009 ; Shell, D. F., et al. (2015). Driver education and teen crashes and traffic violations in the first two years of driving in a graduated licensing system. Accid Anal Prev, 82, 45–52. https://doi.org/10.1016/j.aap.2015.05.011
- Overall, I'm missing the novelty of the work. I do appreciate author's effort in conducting an analysis on such a number of variables, but I lack more interconnections and discussions of the results which could provide some „new“ findings. As such, vast majority of the results, i.e. influential factors are already well known and understood in the literature (please check following review papers: Lin, M. R., Kraus, J. F. (2009). A Review of risk factors and patterns of motorcycle injuries. Accident Analysis & Prevention. 41 (4), 710–722. https://doi.org/10.1016/j.aap.2009.03.010 ; Vlahogianni, E. I., Yannis, G., Golias, J. C. (2012). Overview of critical risk factors in power-two-wheeler safety. Accident Analysis & Prevention. 49, 12– 22. http://dx.doi.org/10.1016/j.aap.2012.04.009; Yousifac, M. T., Sadullaha, A. F. M., Kassim, K. A. A. (2020). A review of behavioural issues contribution to motorcycle safety. IATSS Research. 44 (2), 142-154. https://doi.org/10.1016/j.iatssr.2019.12.001).
- I'm not a native English speaker, but definitely manuscript needs to be checked and proofread by a professional in the field.
Author Response
Please check the attached file for all responses.
Thanks,

Round 2
Reviewer 2 Report
The revision is satisfactory.
Reviewer 3 Report
The authors have addressed the comments. No further comments.
Reviewer 4 Report
Authors addressed all my comments.